# Pollen Application Methods Affecting Fruit Quality and Seed Formation in Artificial Pollination of Yellow-Fleshed Kiwifruit

Eun Ui Oh [1], Seong Cheol Kim [2], Mock Hee Lee [3] and Kwan Jeong Song [1,4,*]

1 Department of Horticultural Science, Jeju National Universtiy, Jeju 63243, Korea; aaa3056@naver.com
2 Research Institute of Climate Change and Agriculture, National Institute of Horticultural & Herbal Science, Rural Development Administration, Jeju 63240, Korea; kimsec@korea.kr
3 Namhae Branch, National Institute of Horticultural & Herbal Science, Rural Development Administration, Namhae 52430, Korea; mockey92@korea.kr
4 Research Institute for Subtropical Agriculture & Biotechnology, Jeju National Universtiy, Jeju 63243, Korea
* Correspondence: kwansong@jejunu.ac.kr; Tel.: +064-754-3328

**Abstract:** This study investigated pollen application methods for artificial pollination in tetraploid kiwifruit cultivars 'Halla Gold' and 'Sweet Gold' grown in a nonheated plastic-film house in Jeju, Korea. Pollen of the hexaploid cultivar 'Bohwa' (*A. chinensis* var. *deliciosa*) bred in Korea was used for artificial pollination. We examined the effect of repeated pollination, pretreatment of stigma with wetting materials, application of dry pollen or pollen in suspension on fruit quality, and seed formation. With repeated pollination, pollen tubes in the pistil reached and penetrated the ovule three days after artificial pollination, although the pattern varied depending on the number of dry pollen applications. In both cultivars, the number of pollen tubes was clearly higher following repeated pollination than following single pollination, and fruit weight, dry matter (DM), number of seeds, and 100-seed weight were also higher. When pistillate flowers were pollinated with dry pollen immediately after water sprinkle, both cultivars showed the lowest fruit weight, DM, firmness, number of seeds, and 100-seed weight, whereas there were no significant differences in fruit quality or seed formation for dry pollen application 1 h after water sprinkle, or immediately or 1 h after suspension medium sprinkle. For pollination using a pollen suspension, the fruit weight was lower in both cultivars. There were no significant differences in fruit quality and seed formation following application of dry pollen or a pollen suspension, except for fruit weight in 'Sweet Gold'. It could be seen from the results of this study that raindrops or dewdrops on the stigma might reduce the efficiency of artificial pollination using dry pollen. Still, the application of repeated pollination enhanced the efficiency of artificial pollination.

**Keywords:** dry pollen; pollen suspension; pollen tube growth; repeated pollination; stigma desiccation

## 1. Introduction

Satisfactory pollination of fruit crops is a key requirement to achieve a sufficient fruit set [1]. Inadequate pollination is one of the main causative factors of low yield and low quality in many fruit tree species, such as olive [2], kiwifruit [3], and pistachio [4]. Kiwifruit (*Actinidia chinensis*) is dioecious, requiring either natural pollination by insects or artificial pollination by humans. For successful natural pollination, the transfer of pollen from staminate flowers to pistillate flowers requires appropriate climatic conditions during the flowering season [5]. Due to the short effective pollination time in kiwifruit [6], growers tend to prefer artificial pollination rather than natural pollination by honeybees to ensure a satisfactory yield [7]. The methods employed for artificial pollination include hand pollination, blowing or spraying of pollen dust, and spraying of pollen suspension [3,5,8–10]. In Korea, artificial pollination of kiwifruit is routinely performed by applying a mixture of dry pollen and lycopodium powder to pistillate flowers, which minimizes management costs.

High-quality pollen is essential for artificial pollination. Recently, many studies on the effect of the cultivar and ploidy of pollen used in artificial pollination on seed formation and fruit quality have been actively reported [11–16]. In addition, other factors also affect fertilization and pollination efficiency, including physiological conditions of the receptive stigma, application time and method, and environmental conditions. Generally, artificial pollination for kiwifruit is performed on a fine day; however, the exudates on the stigma tend to become desiccated around noon when the air temperature is high and humidity low, and this can cause a decline in pollen germination and pollen tube penetration. During artificial pollination, achieving a uniform pollen load on the stigma depends on labor expertise; insufficient or uneven pollen loading by unskillful labor may necessitate repeated pollination. It was suggested in kiwifruit that successive pollinations might arrest the growth of some pollen tubes already in transit through the style by competition with each other [10]. As the transmitting tissue of style has a role of guiding and nurturing pollen tubes, pollen tube competition might be associated with the inhibition of nutrient absorption for already traversed pollen tubes [17]. However, to date, few studies have reported on repeated pollination in kiwifruit, and the procedure is not fully understood. Additionally, kiwifruit producers face many difficulties due to the increase in labor cost and the rising pollen price for artificial pollination. Methods to improve the efficiency of artificial pollination in the kiwifruit industry are still sought. Recent studies have investigated the application of pollen in suspension [18–20], but only a few have involved yellow-fleshed kiwifruit. Several studies have examined pollination methods using the *A. deliciosa* cultivar 'Hayward' [9,17,21,22], but studies on artificial pollination techniques in other cultivars are lacking. This study investigated different pollen application methods for their effect on fruit set and quality and seed formation in two yellow-fleshed kiwifruit cultivars in Jeju, Korea.

## 2. Materials and Methods

### 2.1. Plant Materials

The experiment was carried out for two consecutive years, from 2018 to 2019, on mature vines of the tetraploid pistillate *A. chinensis* var. *chinensis* cultivars 'Halla Gold' (8 years old) and 'Sweet Gold' (5 years old). The kiwifruit vines were trained to the pergola system and cultivated following general management practices such as pruning, fruit thinning, fertilizer management, and pest control in an unheated plastic-film house at commercial fields located in Jeju, Korea.

### 2.2. Pollen Preparation, Viability Testing, and Pollination

Pollen of the hexaploid cultivar 'Bohwa' (*A. chinensis* var. *deliciosa*) bred in Korea was used for artificial pollination. Staminate flowers were collected in both 2017 and 2018. Each flower's anthers, petals, calyx, and peduncle were separated using separation equipment (SWX-400; Samwoo Engineering, Korea). The separated anthers were then dried in a forced-draft static drier (SWX-6000; Samwoo Engineering) at 25 °C for 24 h. Pollen grains were separately collected from the dried anthers using a pollen collector (SWX-6000; Samwoo Engineering). The collected pollen grains were frozen and stored in the freezer. Pollen viability was assessed using two staining methods [23], fluorescein diacetate and 1% iodine potassium iodide, to confirm that the viability of the fresh pollen was ≥90% before use in both years. For the preparation of pollen dusting, pollen was carefully mixed with lycopodium powder at a ratio of 1:10. Pollen suspension was prepared by suspending 4 g of pollen and dissolving 0.2 g of Food Red No. 2 dye (Oh Jung Commercial Co., Ltd., Seoul, Korea) in 1 L of suspension medium. Pollen dusting was carried out using a spray applicator (PS-100, Jeju Bio Tech Co., Ltd., Jeju, Korea), and spraying of the pollen suspension was carried out with a hand sprayer (Apollo Industrial Co., Ltd., Siheung, Korea).

The pollen application experiments were designed to examine three different treatments: differing repetitions of sprays of pollen dust, pretreatment using different wetting

materials before pollen dusting, and dry and wet pollination. Three vines with similar vine size and vigor were selected in both pistillate cultivars for artificial pollination. When 50% of these vines bloomed, about 200 flowers were pollinated for each treatment. In the first experiment, examining repeated pollination, pollen dusting was performed as follows: a single dusting at 10 am on the first day of flowering (M); single dustings at 10 am and 4 pm on the first day of flowering (MA); single dustings in the mornings of the first and second days of flowering (MM); a single dusting in the morning and afternoon of the first day and the morning of the second day (MAM). In the second experiment, examining pretreatment with different wetting materials, pollen was applied on the morning of the second day of flowering as follows: pollen dusting spray without pretreatment; water sprinkling (WS) followed immediately by pollen dusting spray (WS0), WS then pollen dusting spray 1 h later (WS1); suspension medium spray (SS) followed immediately by pollen dusting spray (SS0); SS then pollen dusting spray 1 h later (SS1). The third experiment involved two application types, pollen dusting and pollen suspension, applied on the morning of the second day of flowering.

### 2.3. Observation of Pollen Tube Growth

To observe pollen tube growth after repeated pollination, pistils were collected on the first and third day after final pollination in 2018 and 2019, fixed in formalin/acetic acid/alcohol solution, and kept at 4 °C until further processing [24]. The fixed pistils were softened in 2N NaOH solution at 60 °C for 60–90 min and stained with 0.1% aniline blue for 24 h in the dark at room temperature [25]. The stained pistils were mounted into a block of 4% agarose (Agarose LE, Biomedic Co., Bucheon, Korea), sectioned onto glass slides at a thickness of 7 μm using a Vibratome (Series 1000, The Vibratome Co., St. Louis, MO, USA), and observed under a fluorescence microscope (Leica DMRBE, Leica Co., Wetzlar, Germany) [12].

### 2.4. Fruit Quality Analysis

Fruit quality characteristics were evaluated at harvest using 30 kiwifruits for each treatment. The fruit weight, dry matter (DM), soluble solids content (SSC), acidity, firmness, and flesh color were measured. The fruit weight was measured immediately after harvest, and the DM was measured after drying the equatorial part of the fruit (section thickness 2–3 mm) at 60 °C for 24 h [26]. The SSC and acidity of the juice squeezed from the fruits were measured using a digital sugar and acid analyzer (GMK-707R, G-won Co., Seoul, Korea). Fruit firmness (FF) was measured using a Fruit Texture Analyzer (capacity: 5 kg; FHM-5, Takemura Co., Tokyo, Japan) with a 5-mm diameter plunger after removing the peel and external flesh of fruits to a thickness of 1 mm. The flesh color was measured using a colorimeter (CR-400 Chroma Meter, Minolta Co., Tokyo, Japan) after removing the skin and external flesh to a thickness of 2–3 mm.

### 2.5. Seed Number and Weight

When the harvesting season started, 20 kiwifruits from each of three vines for each treatment were harvested, their seeds were carefully separated, and they were then counted using a seed counter (Countador, Pfeuffer GmbH Co., Kitzingen, Germany). The 100-seed weight was measured using an electronic scale (EL-2000S, Setra Inc., Acton, MA, USA).

### 2.6. Statistical Analysis

Statistical analysis was performed using the SPSS program (SPSS version 18, IBM SPSS Software Inc., Chicago, IL, USA). Significance at the 95% level was tested, and then the significance between the means was analyzed with Duncan's multiple range test.

## 3. Results and Discussion

### 3.1. Repeated Pollination with Dry Pollen

The fertilization process following repeated artificial pollination was investigated by examining pollen tube growth in the pistil (Figure 1). In both 'Halla Gold' and 'Sweet Gold' cultivars, in all treatments (M, MA, MM, and MAM), pollen tubes could be seen to extend past the stigma and along the style one day after pollination (DAP). By three DAP, the number of pollen tubes in the style had increased, and the pollen tubes had reached the ovule in all treatments. The previous study reported that pollen tubes started to reach the ovule on the third day in 'Sweet Gold' and 'Halla Gold' [11], which is in agreement with our observations. However, in our study, the number of pollen tubes reaching the ovule was clearly higher in treatment groups where pollination had been repeated than in group M. This difference was related to the fruit quality parameters and seed formation (Tables 1 and 2).

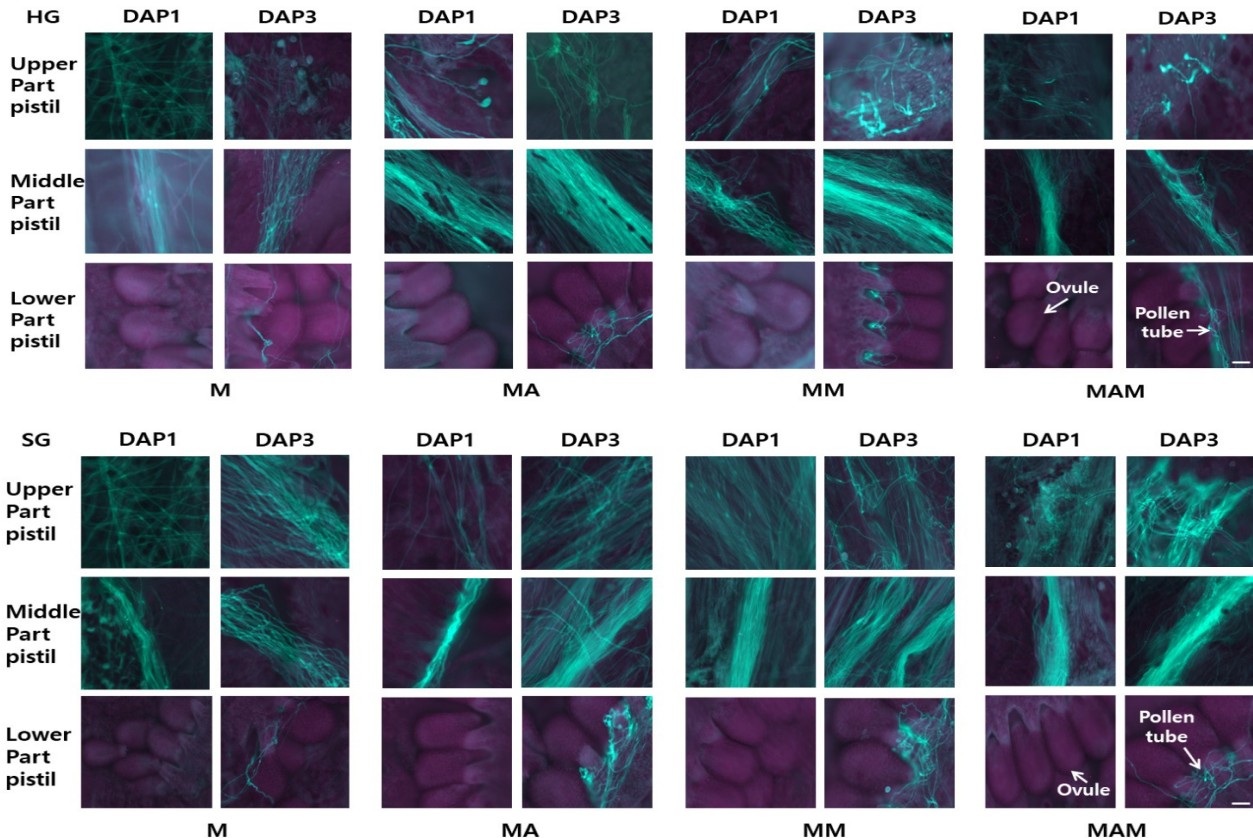

**Figure 1.** Pollen tube growth in the pistils of 'Halla Gold' (HG) and 'Sweet Gold' (SG) pollinated with dry pollen. DAP—days after pollination, M—pollination only in the morning on the day of full bloom, MA—pollination in the morning and afternoon on the day of full bloom, MM—pollination in the morning on the day of full bloom and in the morning on the following day, MAM—pollination in the morning and afternoon on the day of full bloom and in the morning on the following day. Scale bar indicates 100 μm.

**Table 1.** Fruit quality and seed formation in kiwifruit cultivar 'Halla Gold' following repeated pollination with the dry pollen.

| Year | Repeated Pollination | Fruit Weight (g) | Dry Matter (%) | Soluble Solids Content (°Brix) | Acidity (%) | Firmness (kgf) | Flesh Chromaticity $h°$ | Seed Number | 100-Seed Weight (mg) |
|------|---------------------|------------------|----------------|-------------------------------|-------------|----------------|-------------------------|-------------|----------------------|
| 2018 | M [x] (once) | 77.8 ± 0.9 [z] b [y] | 13.0 ± 0.2 bc | 11.8 ± 0.2 | 0.9 ± 0.1 | 5.0 ± 0.1 bc | 98.3 ± 0.7 | 408.0 ± 9.5 b | 12.0 ± 0.02 b |
| | MA (twice) | 82.6 ± 1.1 a | 13.7 ± 0.2 ab | 11.8 ± 0.2 | 0.9 ± 0.1 | 5.6 ± 0.2 ab | 99.9 ± 0.5 | 439.3 ± 11.0 a | 14.1 ± 0.01 a |
| | MM (twice) | 84.3 ± 0.4 a | 14.5 ± 0.3 a | 11.6 ± 0.2 | 0.9 ± 0.1 | 5.5 ± 0.2 a | 99.9 ± 0.5 | 441.6 ± 12.7 a | 14.3 ± 0.01 a |

**Table 1.** *Cont.*

| Year | Repeated Pollination | Fruit Weight (g) | Dry Matter (%) | Soluble Solids Content (°Brix) | Acidity (%) | Firmness (kgf) | Flesh Chromaticity h° | Seed Number | 100-Seed Weight (mg) |
|------|------|------|------|------|------|------|------|------|------|
| | MAM (triple) | 81.0 ± 1.2 ab | 12.3 ± 0.3 c | 11.9 ± 0.2 | 0.9 ± 0.1 | 4.8 ± 0.1 c | 100.5 ± 0.7 | 428.4 ± 7.3 ab | 12.8 ± 0.01 b |
| | Significance [w] | * | * | ns | ns | * | ns | * | * |
| 2019 | M [y] (once) | 91.6 ± 3.3 b | 12.7 ± 0.2 | 11.4 ± 0.2 | 1.2 ± 0.1 | 5.4 ± 0.1 | 108.3 ± 0.4 | 509.7 ± 9.0 b | 12.5 ± 0.01 b |
| | MA (twice) | 95.4 ± 3.2 ab | 13.1 ± 0.2 | 11.3 ± 0.2 | 1.2 ± 0.1 | 5.5 ± 0.1 | 107.3 ± 0.5 | 547.1 ± 5.8 a | 14.9 ± 0.02 a |
| | MM (twice) | 96.9 ± 3.5 a | 12.6 ± 0.3 | 13.5 ± 0.1 | 1.1 ± 0.1 | 5.4 ± 0.1 | 107.9 ± 0.5 | 549.7 ± 8.1 a | 14.4 ± 0.01 a |
| | MAM (triple) | 93.5 ± 3.2 ab | 12.5 ± 0.3 | 11.3 ± 0.2 | 1.2 ± 0.1 | 4.7 ± 0.1 | 107.4 ± 0.5 | 526.1 ± 8.6 ab | 13.1 ± 0.01 ab |
| | Significance | * | ns | ns | ns | ns | ns | * | * |

[z] Mean ± SE (*n* = 30). [y] Different letters in a column within the same species indicate statistically significant differences by Duncan's multiple range test at 5% level. [x] M—pollination only in the morning on the day of full bloom, MA—pollination in morning and afternoon on the day of full bloom, MM—pollination in the morning on the day of full bloom and in the morning on the following day, MAM—pollination in the morning and afternoon on the day of full bloom and in the morning on the following day. [w] ns and *—not significant and significant at *p* < 0.05.

**Table 2.** Fruit quality and seed formation in kiwifruit cultivar 'Sweet Gold' following repeated pollination with the dry pollen.

| Year | Duplicate Pollination | Fruit Weight (g) | Dry Matter (%) | Soluble Solids Content (°Brix) | Acidity (%) | Firmness (kgf) | Flesh Chromaticity h° | Seed Number | 100-Seed Weight (mg) |
|------|------|------|------|------|------|------|------|------|------|
| 2018 | M [x] (once) | 92.7 ± 1.2 [z] c [y] | 17.1 ± 0.1 bc | 12.4 ± 0.1 | 1.0 ± 0.1 | 5.6 ± 0.1 a | 110.0 ± 0.7 a | 808.2 ± 8.5 b | 11.8 ± 0.01 b |
| | MA (twice) | 98.0 ± 0.7 a | 18.3 ± 0.3 a | 12.3 ± 0.2 | 1.0 ± 0.1 | 5.4 ± 0.1 a | 110.0 ± 0.7 a | 845.3 ± 5.5 a | 13.2 ± 0.02 a |
| | MM (twice) | 97.5 ± 0.4 ab | 18.1 ± 0.3 ab | 12.3 ± 0.1 | 1.0 ± 0.1 | 5.5 ± 0.1 a | 106.9 ± 1.1 b | 843.6 ± 7.4 ab | 13.6 ± 0.01 a |
| | MAM (triple) | 94.1 ± 0.9 bc | 16.7 ± 0.3 c | 12.7 ± 0.1 | 1.0 ± 0.1 | 4.5 ± 0.1 b | 111.5 ± 0.2 a | 826.4 ± 8.6 ab | 12.5 ± 0.01 a |
| | Significance [w] | * | * | ns | ns | * | * | * | * |
| 2019 | M [y] (once) | 80.4 ± 0.6 c | 17.4 ± 0.2 ab | 13.1 ± 0.1 | 0.9 ± 0.1 | 5.1 ± 0.1 a | 108.5 ± 0.9 | 760.4 ± 7.6 b | 10.9 ± 0.02 b |
| | MA (twice) | 86.4 ± 0.5 a | 17.7 ± 0.2 a | 13.5 ± 0.2 | 1.0 ± 0.1 | 5.4 ± 0.1 a | 107.1 ± 1.1 | 780.4 ± 9.1 a | 13.0 ± 0.01 a |
| | MM (twice) | 87.2 ± 0.5 a | 17.6 ± 0.2 a | 13.3 ± 0.2 | 0.9 ± 0.1 | 5.1 ± 0.1 a | 107.9 ± 1.0 | 790.8 ± 8.2 a | 13.0 ± 0.01 a |
| | MAM (triple) | 83.8 ± 0.8 b | 16.7 ± 0.2 b | 13.2 ± 0.1 | 0.9 ± 0.1 | 4.8 ± 0.1 b | 108.9 ± 0.9 | 785.6 ± 9.7 a | 12.3 ± 0.02 ab |
| | Significance | * | * | ns | ns | * | ns | * | * |

[z] Mean ± SE (*n* = 30). [y] Different letters in a column within the same species indicate statistically significant differences by Duncan's multiple range test at 5% level. [x] M—pollination only in the morning on the day of full bloom, MA—pollination in the morning and afternoon on the day of full bloom, MM—pollination in the morning on the day of full bloom, and the morning on the following day, MAM—pollination in the morning and the afternoon on the day of full bloom and the morning on the following day. [w] ns and *—not significant and significant at *p* < 0.05.

Fruit weight was highest in treatment MM, and lowest in treatment M. DM was more elevated in treatments MA and MM, while it was typically lower for M and MAM. Values for SSC, acidity, and *h°* did not differ significantly. FF was more deficient in treatment MAM. The number of seeds and 100-seed weight was higher in treatments with repeated pollination than in treatment M. To maximize fertilization, pistillate flowers in most pollination-dependent plants must receive sufficient pollen [27,28]. In 'Hayward' kiwifruit, it has been reported that repeated hand pollination of the same flower with dry pollen over one or more days resulted in smaller fruits with fewer seeds than when single pollination occurs; this was attributed to the arrest or inhibition of pollen tube growth caused by following pollen tubes [10]. However, those results are not reflected in our study in which fruit weight, DM, FF, seed weight, and 100-seed weight appeared to be high when pollen was repeatedly applied to the pistil. On the other hand, it was reported that the number of pollen tubes had an effect on the increase in fruit weight and the number of seeds in 'Hayward' [29]. In this study, it is considered that fruit weight and the number of seeds were high due to the effect of the number of pollen tubes even in repeated fertilization. These differences might be due to differences in cultivar, polyploidy levels, or applied pollen type. Further studies on repeated artificial pollination are needed to elucidate the factors controlling these differences.

*3.2. Pretreatment of Stigma with Wetting Materials*

Results for fruit quality parameters and seed formation following artificial pollination after pretreatment with water or pollen suspension are shown in Table 3 ('Halla Gold') and Table 4 ('Sweet Gold'). Fruit weight, DM, SSC, and FF were the lowest in both years in treatment WS0. Likewise, the number of seeds and 100-seed weight were lowest in treatment WS0. Fruit acidity did not differ significantly. Values for *h°* were typically higher

(except in 2019 for 'Sweet Gold') when artificial pollination was performed immediately after the pretreatment of water (WS0). In light of the observation that treatment with the suspension led to better fruit quality than that with water, it is considered that various nutrients (carbohydrates, proteins, vitamins, minerals, and others) in the suspension had a favorable effect [19].

**Table 3.** Fruit quality and seed formation in kiwifruit cultivar 'Halla Gold' following either dry pollen application after pretreatment with water spray or application of pollen suspension.

| Year | Mode of Pollination | Fruit Weight (g) | Dry Matter (%) | Soluble Solids Content (°Brix) | Acidity (%) | Firmness (kgf) | Flesh Chromaticity $h°$ | Seed Number | 100-Seed Weight (mg) |
|---|---|---|---|---|---|---|---|---|---|
| 2018 | No pretreatment | 80.6 ± 1.1 [z] a [y] | 13.5 ± 0.2 ab | 11.4 ± 0.2 | 1.0 ± 0.1 | 5.2 ± 0.1 ab | 100.5 ± 0.7 ab | 413.8 ± 8.3 a | 12.0 ± 0.2 a |
| | WS (0) [x] | 71.7 ± 2.4 b | 13.0 ± 0.3 b | 11.2 ± 0.2 | 1.0 ± 0.1 | 5.0 ± 0.1 b | 100.8 ± 0.7 a | 321.1 ± 10.9 b | 6.9 ± 0.02 b |
| | WS (1) | 82.5 ± 0.9 a | 13.7 ± 0.2 ab | 11.8 ± 0.1 | 0.9 ± 0.1 | 5.2 ± 0.1 ab | 99.9 ± 0.5 ab | 404.7 ± 10.2 a | 11.9 ± 0.03 a |
| | SS (0) | 83.0 ± 0.7 a | 14.5 ± 0.3 a | 11.6 ± 0.2 | 0.9 ± 0.1 | 5.5 ± 0.2 ab | 98.1 ± 0.6 b | 403.5 ± 9.9 a | 11.1 ± 0.01 a |
| | SS (1) | 84.7 ± 0.8 a | 13.7 ± 0.3 ab | 11.9 ± 0.2 | 0.9 ± 0.1 | 5.7 ± 0.1 a | 98.8 ± 0.5 ab | 414.5 ± 12.3 a | 12.1 ± 0.02 a |
| | Significance [w] | * | * | ns | ns | * | * | * | * |
| 2019 | No pretreatment | 95.6 ± 3.4 a | 13.0 ± 0.3 bc | 11.5 ± 0.2 ab | 1.0 ± 0.1 | 5.4 ± 0.1 ab | 105.1 ± 0.6 ab | 541.8 ± 9.3 a | 13.6 ± 0.01 a |
| | WS (0) [y] | 82.7 ± 1.8 b | 12.4 ± 0.2 c | 10.8 ± 0.1 c | 1.0 ± 0.1 | 4.9 ± 0.1 c | 107.1 ± 0.5 a | 472.3 ± 9.9 b | 10.7 ± 0.02 b |
| | WS (1) | 95.3 ± 0.8 a | 13.2 ± 0.2 bc | 11.2 ± 0.1 bc | 0.9 ± 0.1 | 5.2 ± 0.1 bc | 100.3 ± 0.3 c | 544.0 ± 9.8 a | 13.8 ± 0.02 a |
| | SS (0) | 95.5 ± 0.7 a | 14.4 ± 0.3 a | 11.7 ± 0.1 ab | 1.0 ± 0.1 | 5.8 ± 0.1 a | 103.5 ± 0.4 b | 528.8 ± 9.0 a | 13.2 ± 0.01 ab |
| | SS (1) | 96.7 ± 0.4 a | 13.8 ± 0.2 ab | 12.0 ± 0.1 a | 0.9 ± 0.1 | 5.6 ± 0.1 ab | 103.3 ± 0.4 b | 553.0 ± 8.5 a | 14.0 ± 0.02 a |
| | Significance | * | * | * | ns | * | * | * | * |

[z] Mean ± SE ($n$ = 30). [y] Different letters in a column within the same species indicate statistically significant differences by Duncan's multiple range test at 5% level. [x] WS—water sprinkling, SS—pollen suspension medium spray, 0—immediate pollen dusting spray, 1—pollen dusting spray 1 h later. [w] ns and *—not significant and significant at $p < 0.05$.

**Table 4.** Fruit quality and seed formation in kiwifruit cultivar 'Sweet Gold' following either dry pollen application after pretreatment with water spray or application of pollen suspension.

| Year | Mode of Pollination | Fruit Weight (g) | Dry Matter (%) | Soluble Solids Content (°Brix) | Acidity (%) | Firmness (kgf) | Flesh Chromaticity $h°$ | Seed Number | 100-Seed Weight (mg) |
|---|---|---|---|---|---|---|---|---|---|
| 2018 | No pretreatment | 94.3 ± 1.2 [z] a [y] | 18.7 ± 0.3 a | 12.6 ± 0.1 a | 0.9 ± 0.1 | 5.5 ± 0.1 a | 109.5 ± 0.2 b | 832.9 ± 11.6 a | 12.0 ± 0.02 a |
| | WS (0) [x] | 88.4 ± 0.8 b | 16.8 ± 0.2 b | 11.9 ± 0.2 b | 1.1 ± 0.1 | 4.8 ± 0.1 b | 111.5 ± 0.2 a | 537.9 ± 13.8 b | 6.5 ± 0.01 b |
| | WS (1) | 95.2 ± 0.8 a | 18.1 ± 0.2 a | 12.3 ± 0.2 a | 1.0 ± 0.1 | 5.4 ± 0.1 a | 111.1 ± 0.2 a | 814.4 ± 11.4 a | 11.1 ± 0.02 a |
| | SS (0) | 96.4 ± 0.5 a | 18.6 ± 0.3 a | 12.5 ± 0.2 a | 1.0 ± 0.1 | 5.3 ± 0.1 a | 110.2 ± 0.1 a | 807.5 ± 12.1 a | 11.7 ± 0.02 a |
| | SS (1) | 95.8 ± 0.7 a | 18.7 ± 0.3 a | 12.7 ± 0.1 a | 1.0 ± 0.1 | 5.3 ± 0.2 a | 109.8 ± 0.7 b | 811.9 ± 10.4 a | 11.1 ± 0.01 a |
| | Significance [w] | * | * | * | ns | * | * | * | * |
| 2019 | No pretreatment | 80.2 ± 0.9 a | 17.4 ± 0.2 b | 13.5 ± 0.2 a | 1.0 ± 0.1 | 5.4 ± 0.1 | 109.8 ± 0.9 | 743.1 ± 7.5 a | 11.6 ± 0.02 a |
| | WS (0) [y] | 73.4 ± 0.8 c | 16.7 ± 0.2 ab | 11.7 ± 0.1 b | 0.9 ± 0.1 | 5.0 ± 0.2 | 107.1 ± 2.4 | 536.2 ± 17.6 b | 6.9 ± 0.01 b |
| | WS (1) | 78.4 ± 1.1 ab | 19.0 ± 0.2 a | 12.3 ± 0.1 ab | 1.0 ± 0.1 | 5.2 ± 0.1 | 110.7 ± 0.2 | 751.4 ± 14.2 a | 11.6 ± 0.01 a |
| | SS (0) | 76.3 ± 0.5 b | 18.5 ± 0.2 ab | 12.3 ± 0.1 ab | 1.0 ± 0.1 | 5.5 ± 0.1 | 110.7 ± 0.2 | 763.1 ± 9.4 a | 11.3 ± 0.02 a |
| | SS (1) | 78.0 ± 0.5 b | 18.6 ± 0.2 ab | 12.3 ± 0.4 ab | 1.0 ± 0.1 | 5.1 ± 0.1 | 109.9 ± 0.1 | 752.4 ± 8.2 a | 11.1 ± 0.01 a |
| | Significance | * | * | * | ns | ns | ns | * | * |

[z] Mean ± SE ($n$ = 30). [y] Different letters in a column within the same species indicate statistically significant differences by Duncan's multiple range test at 5% level. [x] WS—water sprinkling, SS—pollen suspension medium spray, 0—immediate pollen dusting spray, 1—pollen dusting spray 1 h later. [w] ns and *—not significant and significant at $p < 0.05$.

It was reported that pollen adhesion in kiwifruit was lowered when it was delivered with water [30]. Furthermore, it was reported that pollen might lose viability due to osmotic shock when it is suspended directly in water, but that it might maintain viability for about 3 h when suspended in a suspension medium [10]. High temperature and high humidity should be avoided in order to maintain the viability of pollen on the stigma. This study showed that immediate pollination after a water sprinkle caused the viability of pollen to be low due to the high humidity of the pistil. These results are in accord with those from orchards, in which artificial pollination on a sunny day rather than on a rainy day normally results in superior fruit quality and seed formation. In addition, since dew often forms on the pistil at dawn, growers should not work early in the morning for effective artificial pollination. Thus, it is recommended that artificial pollination by pollen dusting should be avoided until raindrops or dewdrops on the stigma have evaporated, although further studies on artificial pollination conditions and proper timing are advised.

### 3.3. Application of Dry Pollen or Pollen in Suspension

Table 5 shows results for fruit quality parameters and seed formation in 'Halla Gold' and 'Sweet Gold' kiwifruit at harvest following pollination with either dry pollen or pollen in suspension. Fruit weight, number of seeds, and 100-seed weight in both cultivars tended to be higher following dry pollen pollination than wet pollination, although a significant

difference was observed only in the fruit weight of 'Sweet Gold'. In all other parameters, no statistically significant differences were seen.

**Table 5.** Fruit quality and seed formation in two kiwifruit cultivars following pollination using either dry pollen dusting or pollen in suspension.

| Cultivar | Year | Mode of Pollination | Fresh Weight (g) | Dry Matter (%) | Soluble Solids Content (°Brix) | Acidity (%) | Firmness (kgf) | Flesh Chromaticity $h°$ | Seed Number | 100-Seeds Weight (mg) |
|---|---|---|---|---|---|---|---|---|---|---|
| Halla Gold | 2018 | Dry pollen | 80.6 ± 1.1 [z] | 13.5 ± 0.2 | 11.4 ± 0.2 | 1.0 ± 0.1 | 5.2 ± 0.1 | 98.8 ± 0.5 | 413.8 ± 8.3 | 12.0 |
| | | Wet pollen | 76.6 ± 1.9 | 13.7 ± 0.2 | 12.9 ± 0.2 | 0.9 ± 0.1 | 5.2 ± 0.2 | 99.1 ± 0.6 | 397.9 ± 9.1 | 10.1 |
| | | Significance | ns [y] | ns | ns | ns | ns | ns | ns | ns |
| | 2019 | Dry pollen | 95.6 ± 3.4 | 13.0 ± 0.3 | 11.5 ± 0.2 | 1.2 ± 0.1 | 5.4 ± 0.1 | 107.1 ± 0.5 | 541.8 ± 9.3 | 12.6 |
| | | Wet pollen | 92.1 ± 3.3 | 13.3 ± 0.3 | 12.4 ± 0.2 | 1.2 ± 0.1 | 5.6 ± 0.1 | 108.6 ± 0.5 | 529.1 ± 8.4 | 12.2 |
| | | Significance | ns | ns | ns | ns | ns | ns | ns | ns |
| Sweet Gold | 2018 | Dry pollen | 94.3 ± 1.2 | 17.7 ± 0.3 | 12.6 ± 0.1 | 0.9 ± 0.1 | 5.5 ± 0.1 | 111.1 ± 0.2 | 831.9 ± 11.6 | 12.0 |
| | | Wet pollen | 89.1 ± 1.8 | 18.3 ± 0.3 | 12.7 ± 0.2 | 1.0 ± 0.1 | 5.3 ± 0.2 | 111.5 ± 0.2 | 810.1 ± 9.7 | 11.1 |
| | | Significance [w] | * | ns | ns | ns | ns | ns | ns | ns |
| | 2019 | Dry pollen | 82.1 ± 0.9 | 17.5 ± 0.2 | 13.6 ± 0.2 | 1.0 ± 0.1 | 5.4 ± 0.1 | 109.7 ± 0.9 | 744.1 ± 7.5 | 11.6 |
| | | Wet pollen | 77.4 ± 1.0 | 18.3 ± 0.2 | 13.0 ± 0.2 | 1.0 ± 0.1 | 5.1 ± 0.1 | 107.9 ± 1.0 | 733.7 ± 12.7 | 11.7 |
| | | Significance | * | ns | ns | ns | ns | ns | ns | ns |

[z] Mean ± SE ($n = 30$). [y] Mean separation within columns by *t*-test at 5% level. [w] ns and *—not significant and significant at $p < 0.05$.

It was reported that wet pollen application using pollen suspension and dry pollen application using lycopodium powders did not result in significant differences in the number of seeds, SSC, acidity, or FF in the kiwifruit cultivar 'Hayward' [31], which was in accord with ours. Consequently, we conclude that the use of pollen suspension instead of lycopodium powder as the pollen diluent during pollination of 'Halla Gold' and 'Sweet Gold' cultivars makes no difference to either fruit quality or seed formation. Further studies on suspensions containing nutrients that can enhance the moisture effect on yellow-fleshed kiwifruit are needed [19]. However, using a wet application (pollen suspension) may help reduce operating expenses in orchards since labor and pollen requirements are smaller, though further studies on the appropriate dilution ratio of pollen in the suspension according to the cultivar are needed.

**Author Contributions:** Conceptualization, S.C.K. and K.J.S.; formal analysis, E.U.O. and M.H.L.; funding acquisition, K.J.S.; investigation, E.U.O. and K.J.S.; methodology, E.U.O., S.C.K. and M.H.L.; project administration, K.J.S.; resources, E.U.O., S.C.K. and M.H.L.; supervision, K.J.S.; validation, S.C.K. and M.H.L.; visualization, S.C.K.; writing—original draft, E.U.O.; writing—review & editing, K.J.S. All authors have read and agreed to the published version of the manuscript.

**Funding:** This study was funded by the Collaborative Project Grant (PJ01090402) from the Rural Development Administration.

**Institutional Review Board Statement:** Not applicable.

**Informed Consent Statement:** Not applicable.

**Data Availability Statement:** All data generated or analyzed during this study are included in this published article.

**Conflicts of Interest:** The authors declare no conflict of interest.

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
