# Peer review of "Pollen Application Methods Affecting Fruit Quality and Seed Formation in Artificial Pollination of Yellow-Fleshed Kiwifruit"

_horticulturae, doi:10.3390/horticulturae8020150_

Round 1

Reviewer 1 Report

This is an interesting study as it addresses a key problem and a major concern of kiwifruit producers, particularly in yellow kiwifruit. However, the authors fail to highlight the relevance of their study in this context and to link their results to potential solutions to improve kiwifruit production.  Please see below for further comments.

Abstract

The abstract could be improved. I suggest that the authors start by presenting the context of their study and why it is relevant. The part about the results is too long and could be reduced to the key findings of the study. On the contrary, regarding the methods more details on methods used should be provided, particularly a summary of pollination treatments and pre-treatments, so that reader understands what was done just from reading the abstract.

Finally, the last sentence is confusing and needs rephrasing. Repeated pollination and application type have a minor on what? the efficiency of artificial pollination?  

Introduction

It should be clearer from the introduction why your study is relevant and why/how evaluating different artificial pollination methods my contribute to improve yellow kiwifruit production.

Methods

Overall, methods description needs to be more detailed and informative regarding the experiments performed and treatments within, sample sizes and statistical methods used to analyse the data.

When you first describe the experiment in line 91, it is not clear at all how many experiments, which treatments exist in each experiment or how many replicates per treatment. This should all be better described. Also be clearer as to the timing of pollen application. Was it the first and second day, at the beginning of flowering /peak of flowering?

In statistical analyses, please specify the tests used and what kind of comparisons were made.

Results and discussion

Results and discission section is mostly description of results, there is little discussion and this needs to be improved.

Table captions should be more informative. What are the values provided? Mean  +- SE ? other? This should be included in the table captions

The purpose of the experiment “treating the stigmas with wetting material” is not clear. Why is it relevant, what are the outcomes and how could this be translated into information that could be provided to producers to help them improve their practices. Outcomes and how they can be translated into relevant information to producers is also applicable to the other experiments in the manuscript. Knowledge of when to apply pollen, how many times and quantity are key issues for producers.

L56-58 This statement needs to be supported by references. Please add references to works that support your claims.

L67-68 Please add references to works that support your claims and indicate to which studies you refer.

L109 The sample size for pistil observations is missing. How many pistils/flowers per treatment?

L118 How were you able to recognize the fruits from each treatment? Were the flowers/branches marked differently for each treatment?

L130 Isn't 180 around the harvest time? Why specify 180 if above the authors say harvest time'

L145 Which previous study? Please rephrase the sentence so that it is clear to which study and context you are referring.

L147 Do you mean on the third day? Please clarify.

L149 I suggest using “This difference affected” instead of “This difference is considered to have affected”

L158-172 If the authors intend to stand out the highest and lowest values for each variable they should use the words highest instead of high and lowest instead of low, as they did in section 3.2. Alternatively, authors can use higher in treatment A than in treatment B.  These highs and low, are they statistically significantly different?

L169-70 Please elaborate on why you consider this a possible explanation and provide references to support your statements.

L204-05 This sentence does not make sense. Please clarify and rephrase.

Author Response

Thank you for giving your kind comments and recommendations on our manuscript with your careful review. Please see the attachment. 

Reviewer 2 Report

The reviewed manuscript discusses an interesting and important aspect of kiwifruit cultivation especially since it concerns different cultivars (yellow-fleshed A. chinensis var chinensis - female and A. chinensis var deliciosa - male). Recognizing the different techniques of artificial pollination and identifying the optimal one can be of practical importance for fruit producers. An additional aspect is their different ploidy, which may have different consequences for fruit production.   

However, in my opinion the manuscript require significant improvement before is ready for publication. Below are my comments on the individual sections of the manuscript:

Introduction: The Authors omitted the aspect of the influence of ploidy on the pollination process in Actinidia. As yellow-fleshed kiwifruit are usually of different ploidy level than green-fleshed it is important to review ploidy impact on pollination process using pollen of different ploidy. I suggest mentioning a few recent publications on this topic. This will allow the authors to discuss the obtained results a bit more extensively. Especially that the authors conducted such research and published their results in 2021 (https://doi.org/10.1007/s13580-020-00293-z).

Material and Methods: Should be explained how was pollen tube overgrow assessed? Did you used (or developed) any methodology? More detailed information is needed. It should be clearly stated how many plants was used as repetition for each of three experiments performed. How was pollen collected? The most A. chinensis var. chinensis genotypes flowers earlier than A. chinensis var. deliciosa. Was pollen collected and used in the same season? How was pollen stored until the experiment? These aspects should be described in more detail.

Results and Discussion:  I suggest discussing the reaction of both cultivars and the influence of the research season on particular aspects in more detail. Was the reaction the same in both seasons? The discussion of the results is too briefly and does not include recent publications (e.g. the authors' publication indicated above or Broussard et al. 2021 https://doi.org/10.1080/01140671.2020.1861032; Stasiak et al. 2021 https://doi.org/10.3390/agronomy11091814 or Abbate et al. 2021 https://doi.org/10.1093/jee/toab075).

Some other editorial remarks are indicated in the text attached.

Author Response

(The authors gave the same response as above.)

Round 2

Reviewer 1 Report

The authors have made changes according to the earlier review. I detected a minor English error in L63. It should read “competition with each other”.

Author Response

Thank you for your careful review. In L63, the phrase was changed with the reviewer's indication, “competition with each other”.

Reviewer 2 Report

As the authors took most of my suggestions into account, I accept the article in it's current form. Minor correction suggestion - Line 52 instead of "polyploidy" insert "ploidy".

Author Response

Thank you for your careful review.  The word of polyploidy was changed with ploidy as reviewer recommended.